# Biomarkers in Hepatocellular Carcinoma: Diagnosis, Prognosis and Treatment Response Assessment

**DOI:** 10.3390/cells9061370

**Published:** 2020-06-01

**Authors:** Federico Piñero, Melisa Dirchwolf, Mário G. Pessôa

**Affiliations:** 1Hepatology and Liver Transplant Unit, Hospital Universitario Austral, School of Medicine, Austral University, B1629AHJ Buenos Aires, Argentina; fpinerof@cas.austral.edu.ar; 2Latin American Liver Research Educational and Awareness Network (LALREAN), B1629AHJ Buenos Aires, Argentina; 3Liver Unit, Hospital Privado de Rosario, 2000 Rosario, Santa Fe, Argentina; mdirchwolf@outlook.com; 4Division of Gastroenterology and Hepatology, University of São Paulo School of Medicine, 05403-000 São Paulo, Brazil

**Keywords:** liver cancer, biological, markers

## Abstract

Hepatocellular carcinoma (HCC) is one of the main cancer-related causes of death worldwide. Thus, there is a constant search for improvement in screening, diagnosis, and treatment strategies to improve the prognosis of this malignancy. The identification of useful biomarkers for surveillance and early HCC diagnosis is still deficient, with available serum biomarkers showing low sensitivity and heterogeneous specificity despite different cut-off points, even when assessed longitudinally, or with a combination of serum biomarkers. In contrast, HCC biomarkers used for prognostic (when associated with clinical outcomes) or predictive purposes (when associated with treatment response) may have an increased clinical role in the near future. Furthermore, some serum biomarkers are already implicated as a treatment selection tool, whether to provide access to certain therapies or to assess clinical benefit after treatment. In the present review we will discuss the clinical utility and foreseen future of HCC biomarkers implicated in surveillance, diagnosis, prognosis, and post-treatment assessment.

## 1. Introduction

Hepatocellular carcinoma (HCC) is nowadays one of the most frequent malignancies and a leading cancer-related cause of death worldwide [1,2]. During the last decades, some improvements in the therapeutic approach have been achieved not only for early but also for advanced HCC stages. On the contrary, there has not been a significant clinical improvement in HCC biomarkers for surveillance and early diagnosis. On the other hand, for prognosis and treatment response purposes, these biomarkers might have a clinical role.

Although different serum or tissue biomarkers have already been studied, their clinical utility has not been widely accepted. One of the most important issues, and maybe their Achilles heel, is their low sensitivity (with high false-negative results), opposed to high specificity that precludes HCC biomarkers to be clinically useful for early HCC diagnosis. Moreover, there is a huge amount of publications with different and heterogeneous cut-offs with corresponding sensitivities and specificities.

However, most of these biomarkers have been associated with poor prognosis, either in early or advanced HCC. Besides, tumor markers for appropriate treatment selection or response have been widely evaluated in recent years. Even with the advent of new therapeutic oncologic modalities, such as immunotherapy with checkpoint inhibitors [3,4,5], new biomarkers have not settled into daily practice, except for alpha-fetoprotein (AFP) [6]. Therefore, the challenge is to develop other biomarkers for early diagnosis, adequate treatment selection of patients, and post-treatment prognosis, including proteomics, metabolomics, genomics, and other novel biomarkers such as microbiome [7,8,9,10].

In this review, we describe HCC serum and tissue biomarkers focusing on their clinical utility upon HCC surveillance, early diagnosis, prognosis, and post-treatment assessment.

## 2. Biomarkers for Hepatocellular Carcinoma Surveillance and Diagnosis

### 2.1. The Utility of Serum Biomarkers for Hepatocellular Carcinoma (HCC) Surveillance

Although the evidence-based data for HCC surveillance is low to moderate, including only two randomized trials in patients with chronic hepatitis B infection (HBV) with a significant risk of bias [11,12] and several observational studies in cirrhosis [13,14,15,16], surveillance for HCC is broadly recommended by international guidelines [17]. This recommendation is supported by several epidemiological reasons. First of all, HCC represents an important public health concern, currently being the fourth cancer-related cause of death worldwide [2]. It has a well-defined population at risk, including patients with chronic hepatitis B (HBV) or C virus (HCV), and patients with any chronic liver disease with severe fibrosis or cirrhosis. These group of patients have an estimated cumulative incidence of HCC above 1.5% per year [18]. Furthermore, due to its asymptomatic pre-clinical stage, early diagnosis of HCC is feasible, providing higher access to curative treatments, and significant improvement in overall survival [19].

It should be noted that the evidence regarding HCC surveillance is moderate in chronic HBV and weak in patients with cirrhosis [11,12,13,14,15,16]. No randomized controlled trials assessing the survival benefit of HCC surveillance in patients with cirrhosis were reported, and in some studies, factors such as lead-time bias (apparent improvement in survival derived from early diagnosis) were not appropriately considered [20]. However, due to the ethical conflicts of conducting a trial with a non-interventional arm, current recommendations are based on the available evidence provided by cohort studies with a significant risk of bias.

Finally, for a surveillance program to be successful, the screening test should be easily available, cheap, reproducible, and with an appropriate detection accuracy [21]. Most scientific societies endorse a 6-month interval evaluation with abdominal ultrasonography (US) [22]. The main drawback is the fact that it relies on an operator-dependent method, thus offering heterogeneous results according to the expertise of the imaging specialist [23]. Moreover, the potential benefits and harms of HCC surveillance have been described with a 25% rate of false-positive results [24]. Asian countries have proposed to include biomarkers other than AFP but in Western countries these were not included in their recommendations [25,26,27]. Consequently, no international consensus has been reached so far regarding the ideal biomarker to be used as a surveillance tool.

### 2.2. Difficulties Related to the Validation of Serum Biomarkers for HCC Surveillance

#### 2.2.1. Alpha-Fetoprotein (AFP)

Alpha-fetoprotein (AFP) is the most commonly used biomarker for HCC surveillance. This 70 kD glycoprotein is produced by the fetal liver and yolk sac during the first trimester of pregnancy and declines rapidly after birth [28]. AFP is structurally very similar to albumin, with just a modification in an N terminal sequence. It was first described as a useful biomarker for HCC over fifty years ago, in murine models, and later in African and Siberian population studies [29]. Since its introduction as a screening tool for HCC, its utility has been challenged.

##### The Specificity of Alpha-Fetoprotein

Serum AFP can be elevated in other benign or malignant conditions. Its specificity is undermined by its elevation in other conditions such as acute and chronic hepatitis, intrahepatic cholangiocarcinoma, and embryogenic tumors [29]. Particularly, elevated AFP levels in patients with chronic HCV renders this biomarker conflictive for HCC screening purposes in this population (associated with necro-inflammatory activity). In the HALT-C trial that included over 1000 HCV+ patients with moderate to severe fibrosis [30], 11% and 27% of patients with bridging fibrosis or cirrhosis had AFP values above 20 ng/mL, respectively [31]. Interestingly, only six patients developed HCC, and only three of them had elevated AFP values [31]. Besides, at least 25% of the patients had AFP values above 20 ng/mL at least once during follow-up without HCC development [32]. More recently, the lack of reduction of AFP values during HCV treatment with direct-acting antivirals has been proposed as an independent risk factor for HCC development; this approach is yet to be validated [33]. Finally, it should be clarified that AFP values have been reported in ng/mL or IU/mL. Conversion from IU to ng should be done multiplying IU for 0.05 (e.g., 10 IU/mL is equivalent to 10.5 ng/mL).

##### AFP Cut-Offs for Surveillance and HCC Diagnosis

The role of AFP as a screening tool and the selection of its most accurate cutoff depends on the prevalence of HCC in the tested population. Sensitivities and specificities of AFP values vary according to which cut-off for HCC early diagnosis is selected [34]. For a cutoff value of 20 ng/mL, in a population with a 5% prevalence of HCC, Trevisani et al. found a negative predictive value (NPV) of 97.7%, and a positive predictive value (PPV) of 25% [34]. Thus, rendering this biomarker useful for exclusion of HCC but poor for its early detection [34]. In comparison, when an HCC prevalence of 20% was considered, this cutoff had a PPV of 61% maintaining a NPV of 90% [34]. Consequently, the AFP cut-off with the highest sensitivity for early HCC detection was of >20 ng/mL (60%) when compared to other cut-offs (>100 ng/mL, >200 ng/mL and >400 ng/mL) [34]. On the contrary, specificities increased with increasing cut-off values. Thus, false negative and false positive rates vary accordingly [34] (Figure 1). These heterogeneous results were observed when AFP was used as a screening tool in different geographic areas, favoring its use in Eastern populations and poorly performing in Western populations [32,35]. Racial disparities have also been reported [36].

Two meta-analyses evaluated the utility of adding AFP to abdominal US for HCC surveillance [37,38]. The first study included 13 studies and described no significant differences in sensitivities for HCC detection at early stages with US alone (63%) and combined with AFP (69%) [37]. In contrast, a recent meta-analysis including 32 studies found that US with vs. without AFP presented a significantly higher sensitivity for early-stage HCC (63% vs. 45%; *P* = 0.002) [38]. It should be noted that the later meta-analysis observed a wide heterogeneity in ultrasound performance that could not be fully explained by subgroup analysis [38].

These opposed results are reflected in current major guidelines recommendations. The Asia-Pacific Association for the Study of the Liver (APASL) guidelines endorse its use in combination with abdominal US [22], whereas the European Association for the Study of the Liver (EASL) guidelines state that all tested biomarkers (including AFP) are suboptimal in terms of cost-effectiveness [25], and finally, the American Association for the Study of Liver Diseases (AASLD) mention both surveillance strategies (ultrasonography with or without AFP) as equivalent [26].

#### 2.2.2. Suggested Approaches to Improve AFP Accuracy in Surveillance

##### Risk Stratification for Surveillance Algorithm

Another novel approach is to stratify the risk of HCC development, based on additional scoring models. In HBV, the PAGE score has been developed in Caucasians and validated in Asian population [39]. This score can identify patients with chronic HBV that are at higher risk of HCC and require continuous HCC surveillance even after treatment with HBV antivirals. Among HCV patients, other scoring models to assess the risk of HCC have been addressed including AFP values, age, platelets count, and ALT levels [40,41].

##### Longitudinal Changes on Serum AFP

When the performance of dynamic changes of AFP was tested in a population with an HCC prevalence of 3%, the combination of baseline AFP >10 ng/mL in combination with increasing AFP levels increased the sensitivity to 80% with a NPV of 99% [42]. This encouraging approach is yet to be validated. Another longitudinal approach based on Bayesian modeling was proposed based on the HALT-C trial [43]. This modeling approach was proposed in patients with HCV considering different cut-offs and longitudinal AFP changes during follow-up. It is still a matter of debate, whether an AFP specific cut-off (>200 ng/mL) adds any clinical additional tool to detect HCC at an early stage in patients with a negative US test.

##### Longitudinal Assessment of Combined Serum Biomarkers

Another recent novel approach is the sequential and longitudinal evaluation of combined HCC biomarkers during a 12-month follow-up period [44]. In a case-control study nested on a prospective observational cohort and in 3 randomized clinical trials of patients with chronic HBV, the longitudinal assessment of AFP, DCP, and AFP-L3 were compared [44]. AFP had the highest AUROC for early HCC diagnosis when compared to AFP-L3 (highly sensitive assay), and DCP. The combination of AFP (cut-off >5 ng/mL) and AFP-L3 (cut-off >4%) showed the highest AUROC value (0.83) when compared to any single biomarker [44]. However, while this resulted in an increased sensitivity, a decreased specificity was observed with the combination of biomarkers [44].

Some authors have addressed the risk of developing HCC after HCV viral eradication with clinical variables [45]. The presence of clinically significant portal hypertension is one of the most important independent variables associated with the risk of developing HCC. Thus, among patients with clinically significant portal hypertension, surveillance should be further stricter or underlined. It remains uncertain whether the combination or longitudinal assessment of these biomarkers following HCV eradication may optimize HCC detection rates at early stages, particularly in patients remaining at higher risk of HCC.

#### 2.2.3. Alpha-Fetoprotein Lens Culinaris Agglutin-3 (AFP-L3)

Total AFP can be separated into three fractions, AFP-L1 to AFP-L3, based on its reactivity to *Lens culinaris* agglutinin (LCA) affinity on electrophoresis [28]. Of these three isoforms, the AFP-L3 fraction appears to be more specific for HCC since it is produced exclusively by HCC cells. When initially evaluated as a screening test, an elevated fraction (over 15%) of AFP-L3% showed slightly better sensitivity than AFP (38% vs. 31%) for HCC detection [46]. It’s promising use as a surveillance test was suggested from retrospective studies, in which 95% and 71% of the patients had positive values of AFP-L3% at 3 and 6 months before diagnosis, respectively [46]. However, in further studies its sensitivity and specificity greatly varied when cutoffs of 10–15% were used (from 36–96% and 89–94% respectively) [47]. In a previously published metanalysis comparing AFP, AFP-L3 and Des-γ-carboxy prothrombin (DCP) to detect a single HCC nodule of less than 5 cm, AUROCs were 0.65, 0.69, and 0.69 with corresponding cut-offs of >200 ng/mL, >15% and >40 mAU/mL, respectively [48]. When compared with total AFP, AFP-L3% had a lower AUROC for differentiating cirrhosis from early HCC compared in two case-control studies mainly enrolling non-Hispanic HCV patients [48]. However, in a prospective cohort study from North America, a similar accuracy between AFP, AFP-L3%, and Des-γ-carboxy prothrombin (DCP) was observed [49]. Only when these tests were used in combination, the sensitivity increased to 77% while maintaining a high NPV (91%) for HCC surveillance [50]. Consequently, it seems that the combination of these serum biomarkers might increase sensitivity at the expense of decreasing specificity when compared to each biomarker alone [51].

Some technical issues must be addressed regarding the measurement of APF-L3 fraction. In previously published papers, AFP-L3 was always measured simultaneously with AFP and its significance depended on AFP values. Serum AFP values are usually measured by an immunometric assay, whereas AFP-L3 levels by lectin-affinity electrophoresis coupled with antibody-affinity blotting. It is expressed by the ratio of AFP-L3 to the total AFP in a percentage (%). However, AFP-L 3% cannot be precisely measured in patients with low AFP values (<10 ng/mL) and in some patients with higher AFP values (>400 ng/mL), the AFP-L3 proportion might not be accurately defined [51,52]. Consequently, in many previously published papers, the ratio and AFP-L3% has been artificially ranged between 20 and 200 ng/mL of AFP total values. More recently, another technique was developed through a micro total analysis system (micro-TAS), which has been proposed to be useful in cases with AFP values below 20 ng/mL [53,54]. Thus, the greatest clinical utility of AFP-L3 or DCP has been proposed to be in patients with intermediate AFP values (20–200 ng/mL); in which they have shown to be highly specific for HCC [52].

#### 2.2.4. Des-γ-Carboxy Prothrombin (DCP)

Also known as prothrombin induced by vitamin K absence-II (PIVKA-II), this abnormal protein without coagulant function is presumably caused by an acquired defect in the posttranslational carboxylation of the prothrombin precursor in malignant cells [55]. DCP has been described as a useful tool for HCC surveillance since it is independent of AFP secretion. However, its efficacy as a screening tool is still controversial. In a large-scale Chinese multicenter study evaluating the role of DCP in HBV related HCC, DCP had better accuracy than AFP (88.5% vs. 76.2% respectively) as a surveillance tool with a cutoff level of 40 mAU/mL [56]. Furthermore, the diagnostic accuracy of AFP plus DCP was slightly improved when compared to DCP alone (1.12–2.69%) and significantly improved when compared to AFP (8.26–13.42%) [56]. This biomarker also achieved a high accuracy in AFP-negative HCC patients (AUROC of 0.86) [56]. In contrast, in a Caucasian HCV+ cohort, AFP was found to be more sensitive than DCP for HCC diagnosis at early stages with a cutoff of 10.9 ng/mL [50]. Some authors proposed that the combination of DCP with AFP and/or AFP-L3% may increase the sensitivity and PPV for early HCC, whereas other authors described lower specificities when these biomarkers were combined [55].

Some technical details should be addressed. First of all, DCP is not accurately quantified in the presence of vitamin K deficiency, the use of oral anticoagulants, or in patients with a poor nutritional status associated with alcohol abuse [57,58]. Moreover, in some cases, two different antibodies were needed to accurately measure DCP (P-11 and P-16). The next generation method for DCP quantification was created to avoid this latter technical issue [57,59].

### 2.3. Difficulties Related to the Use of Serum Biomarkers for HCC Diagnosis

The diagnosis of HCC has been settling down as a paradigm in clinical oncology. Through its organ specificity, a non-invasive diagnosis can be done by a three-phase dynamic computed tomography (CT) scan or a magnetic resonance image (MRI) in patients at high risk of HCC [25,26]. Specificity for HCC diagnosis is higher than 90% in these clinical scenarios, with the presence of specific radiological hallmarks as arterial phase hyper-enhancement (APHE) and wash-out during late or portal phases [60,61]. However, some nodules may not present these typical hallmarks or may present with additional non-specific features. Thus, tumor biopsy may be appropriate in these cases [25,26].

In 2008, the American college of radiology proposed the Liver Imaging Reporting and Data System (LI-RADS) to standardize the imaging interpretation and the reporting of nodular liver lesions in patients at high risk of HCC [60,62]. The LI-RADS first version appeared online during 2011 and since then it has been updated [62,63]. It categorizes the probability or likelihood of malignancy. While LI-RADS 2 (LR-2) are more likely to be benign nodules, LI-RADS 3 to 5 have increasing probabilities of HCC. LI-RADS M defines malignancy not specifically HCC [62,63].

The limited use of biomarkers for HCC diagnosis may be explained because during the last decades, imaging technology has widely improved. Moreover, although the combination of serum AFP, AFP-L3 and DCP have been proposed for HCC surveillance in Asia, their clinical use in Western regions have not settled down. In this regard, the combination of these serum biomarkers added a marginal improvement of HCC surveillance or diagnosis [52]. HCC biomarkers are elevated in a low proportion of patients, they are associated with tumor burden and maybe not be useful for diagnosis at early stages. Thus, low sensitivities and not enough specificities have been reported. Consequently, imaging diagnosis is superior to biomarkers in daily practice. Nevertheless, in the context of a clinical suspicion setting (population at risk) with atypical imaging features, the use of biomarkers may be an additional tool for HCC diagnosis.

The poor performance of total AFP in detecting very early HCC has led to a past interest in identifying tumor markers. In Japan, AFP-L3 and DCP in combination with AFP have been proposed [64]. However, AFP-L3 and DCP have not been associated with a higher HCC detection rate and both have been associated with larger tumors, metastatic disease, dedifferentiated tumors, or vascular invasion [65,66]. Besides, elevated AFP-L3% are usually undetectable in patients with lower AFP values (<10 ng/mL) and are usually seen in those with elevated total AFP [48,50,67]. As previously mentioned, the combination of these biomarkers have been associated with increasing sensitivities but decreasing specificities [68]. Consequently, most of the international Western clinical guidelines do not recommend the diagnosis of HCC based on biomarkers [25,26]. In the end, the final aim of tumor biomarkers should promote HCC diagnosis at early stages and a decrease in mortality rates [69,70] (Table 1).

### 2.4. Scoring Models for HCC Surveillance and Early Diagnosis

Several authors have developed predictive models combining patient characteristics with serum biomarkers for HCC surveillance and early detection. The GALAD score was developed including Gender, Age, and a Logarithmic transformation of three biomarkers (***A***FP, AFP-L3, and DCP) [71]. This model showed a very good discrimination power with an AUROC of 0.97 irrespective of etiology and disease stage. Other authors have proposed another scoring model for HCC surveillance, the GALADUS score [71]. The authors evaluated the efficacy of the GALAD score in comparison to abdominal US. The AUROC for the GALAD score (0.95; CI 0.93–0.97) was significantly higher than that of US alone (0.82). This AUROC remained higher for the GALAD score irrespective of disease etiology (US had a lower AUROC in patients with alcoholic liver disease), AFP values, or with the presence of ascites in which the US performance is usually affected. Focusing on HCC detection at early stages, the AUROC for GALAD score was still higher than US alone (0.92 vs. 0.82; *P* < 0.01). The combination of GALAD and US presented even higher AUROC for HCC detection at early stages, particularly for patients with a negative US result. Nevertheless, these results were based on retrospective cohorts with some risk of bias. Although this approach is interesting, it needs to be prospectively validated. The aim of these models should be to detect small HCC single lesions of less than 3 cm in diameter, particularly in those cases with a negative US result.

### 2.5. Other Novel Serum Biomarkers for HCC Diagnosis

Other tumor biomarkers have been proposed such as osteopontin (OPN), vascular endothelial growth factor (VEGF), angiopoietin 2 (ANG-2), Golgi protein 73 (Gp-73), insulin growth factor-1 (IGF-1), hepatic growth factor (HGF), Glypican-3 and c-MET among others. Serum OPN has been associated with increasing AFP serum levels, p53 mutation, vascular invasion, dedifferentiated HCC, and with poor prognosis [72]. However, its accuracy in HCC detection at early tumor stages has not been established. VEGF, AP-2, HGF and c-MET have been proposed as prognostic or predictive markers and will be further detailed. Decreasing levels of IGF-1 in HCV+ cirrhotic patients were associated with HCC development during follow up, independently from liver function [73]. This biomarker has been proposed to identify patients with a preceding HCC diagnosis but has not been implemented in daily practice. The Gp-73 has been assessed in a case-control study including HCC patients, cirrhotic patients without HCC and healthy controls, matched for age, gender, and race [74]. The AUROC outperformed AFP for HCC diagnosis (0.79 vs. 0.61) with a sensitivity and specificity of 69% and 75% for a cut-off value of 10 relative units. The AUROC for HCC detection at an early stage was 0.77 (95%; CI 0.70–0.85), with a sensitivity of 62%. Thus, a 38% false-negative rate precluded this biomarker to be a novel tool for HCC surveillance. Serum dickkopf-1 (DKK1), a secretory antagonist of the Wnt signaling pathway, was highly expressed in HCC tissue and not detectable in the non-tumor liver. In a prospective HBV+ cohort, this serum biomarker was proposed to be a novel HCC biomarker with a very good diagnostic performance for HCC, even in early stages and in patients with normal AFP values (<20 ng/mL) (AUROC of 0.87 ;CI 0.83–0.91) [75].

Glypican-3, a cell-surface heparan sulfate glycoprotein, is highly expressed in HCC. In tumor samples, it has been included for immunohistochemistry assays on pathology specimens. There is no pathognomonic immunohistochemistry for HCC but the presence of at least 2 out of 3 markers, Glypican 3 or Heat Shock Protein 70 or Glutamine sintetase, has 60% sensitivity and 100% specificity for HCC [76]. Besides, Glypican-3 on tissue samples has been associated with poor prognosis in patients with HCC [77,78]. It has been identified as a prognostic marker in two metanalysis. Liu et al. associated Glypican-3 with aggressive histological tumor features, tumor progression, metastasis, and poorer overall survival [78]. However, significant heterogeneity was reported when evaluating overall survival (I^2^ of 65%) and disease-free survival (I^2^ of 81%) [78]. In another metanalysis, similar observations were reported, associating Glypican-3 with aggressive histological tumor features (dedifferentiation and vascular invasion). Higher expression of Glypican-3 on tissue samples was significantly associated with worse overall and disease-free survival [77]. On the other hand, serum Glypican-3 has been evaluated for surveillance and early HCC diagnosis [79]. Its NH2-terminal portion, which is soluble (soluble GPC3), can be detected and quantified in serum samples. However, when compared to AFP, Glypican-3 showed modest accuracy with a lower AUROC compared to AFP at a cut-off value of 20 ng/mL (0.72 vs. 0.80, respectively). Even when assessed in well-differentiated tumors, its AUROC was not significantly better than AFP [79].

Genomic profiling and proteomics have gained popularity and are being focused on novel research for HCC diagnosis during the last years [80,81]. Plasma micro-RNA has been linked to oncogenesis and tumor metastasis. A large number of circulating micro-RNAs have been identified in HCC patients, some of them associated with potential HCC diagnostic or prognostic implications [82,83,84,85]. However, these novel biomarkers have been further tested as prognostic and predictive factors.

## 3. Serum Biomarkers in HCC for Tumor Staging and Prognosis

HCC is a unique tumor that includes not only tumor features but also remnant liver function as key prognostic factors for survival. Thus, liver function, portal hypertension, and its complications act as competing events for survival because most of the patients with HCC have underlying chronic liver disease or cirrhosis [86,87,88]. According to different staging algorithms, biomarkers have or not been included in these clinical staging systems. 

It is important to specifically clarify the clinical implications and definition of prognostic and predictive factors. Although these terms may sound similar, they are different. A prognostic factor is an exposure baseline variable that it is independently associated with a worse clinical outcome. On the contrary, a predictive factor may or may be not be a prognostic factor, but it identifies a population with a better or worse response to a particular treatment.

The BCLC clinical algorithm has been proposed years before based on the most relevant scientific evidence: it provides a rationale for the clinical-decision-making processes [89,90]. The BCLC includes different prognostic clinical and tumor burden variables. Total bilirubin, presence of portal hypertension, preserved liver function (absence of clinically significant portal hypertension, including ascites and its complications) and Eastern Cooperative Oncology Group (ECOG) performance status are associated with prognosis and were included in this algorithm.

However, biomarkers have not been included in the BCLC. Other staging proposals included AFP values for HCC staging and prognosis, such as the Cancer of the Liver Italian Program (CLIP) [91], the GRETCH staging from France [92] and the Chinese University Prognostic Index (CUPI) [93]. The Japan Integrated Scoring system integrates the TNM tumor staging with the Child–Pugh score [94,95]. More recently, another staging system proposed by the Hong Kong Liver Cancer group did not include AFP values or other biomarkers to the clinical-decision-making processes [96] (Table 2).

Other authors proposed to assessed HCC staging excluding imaging or clinical data [97,98]. These new models based on biochemical data with or without biomarkers, have been developed to surpass the subjectivity of scores and grading systems based on clinical data to assess liver function (Child–Pugh score, including ascites or encephalopathy grades). The BALAD score includes 2 biochemical variables (serum bilirubin, albumin) and three biomarkers [97]. This scoring model included a panel of 3 biomarkers (AFP >400 ng/mL, AFP-L3 >15% and DCP >100 mAU/mL) and their combination was associated with worse prognosis [64]. However, as this score does not include any imaging tumor features or clinically relevant data, it has not been widely implemented in daily practice. Moreover, although survival was well assessed, there might have been a treatment selection bias. The discrimination power of BALAD and BALAD-2 scores were not superior to other clinical staging algorithms that did not include biomarkers, as the JIS [99,100]. Nevertheless, the BALAD score was recently externally validated in Asian and Western populations [99,100,101]. Although it was designed for HCC detection rather than prognosis, the BALAD-2 score was found to be associated with prognosis in another external validation cohort [102]. More recently, the GALAD score has been validated in patients with non-alcoholic fatty liver, either with or without cirrhosis [102].

The ALBI grade was developed to avoid any subjective bias interpretation of clinical data, including ascites or other portal hypertension complications [98]. It did not include performance status. Although novel and interesting, this score presented a modest discrimination power with a Harrells’ c-statistic lower than 0.70 in the test and validation cohorts. It did not add any significant gain on discrimination power over the Child–Pugh score [98] (Table 3).

### 3.1. Biomarkers as Clinical Prognostic Tools

#### 3.1.1. Alpha-Fetoprotein

AFP has been widely and extensively studied in association with prognosis. However, controversy has been raised regarding which specific cut-off for survival or recurrence should be chosen. Several cut-offs have been proposed associated with worse survival. Nevertheless, increasing AFP values are associated with lower survival and higher tumor recurrence rate in patients at very early or early stages [103,104,105,106], as well as poor prognosis in patients with advanced HCC [107].

In patients at very or early stages, AFP values above 1000 ng/mL were associated with worse survival and higher recurrence rates [108]. On the other hand, this biomarker has been correlated with microvascular invasion and tumor dedifferentiation on explant pathology analysis after liver transplantation (LT) [103,105]. Consequently, AFP values are useful for selection of appropriate or best candidates for surgical options or LT. In this regard, since 2013, France has adopted the AFP model, which includes AFP values and radiological tumor burden, as a selection tool for transplant candidates [103]. Besides, in the United States of America, a restriction policy for LT has been implemented with AFP serum levels above 1000 ng/mL [109]. Other authors have proposed other cut-offs [110,111] or continuous AFP values [106].

In patients at intermediate stages, AFP has been associated with tumor progression in patients listed for LT who underwent tumor bridging therapies while on the waiting list or in patients who received locoregional tumor treatment to reduce their tumor burden. This latter objective, called *downstaging* from beyond to within tumor transplant limits [112,113]. Serum AFP values above 400 ng/mL have been associated with higher waitlist tumor progression and a lower response rate after trans-arterial chemoembolization (TACE) [114,115]. Moreover, AFP values above 100 ng/mL have been recently proposed as a selection criterion for the best candidates for downstaging [116].

Among patients with advanced HCC, serum AFP values have been associated with worse baseline prognosis in the SHARP and Asia Pacific trials evaluating the efficacy of sorafenib [117,118]. Based on these trials, AFP values above 200 ng/mL were associated with a poor prognosis in both treatment arms; however, sorafenib was effective even in this population [107]. Although AFP was associated with lower survival in this latter group, it was not included as a stratification factor in some trials assessing first line systemic treatment options [119,120,121,122,123]. AFP values have been associated with worse clinical outcomes in patients with tumor progression during sorafenib treatment [124]. Consequently, other trials for second line systemic treatments have included it as a stratification factor [125] or as an exclusion eligibility criteria [5]. Moreover, AFP values above 400 ng/mL have been associated with response criteria for ramucirumab; thus, showing to be the first biomarker used as selection criteria for systemic treatment options [6].

#### 3.1.2. Alpha-Fetoprotein LP-3

AFP-L3 has been widely and extensively studied. This biomarker has been associated with biologically more aggressive tumors too. Originally described to be associated with multiple HCC recurrences, vascular tumor invasion, dedifferentiated tumors, and poorer overall survival [65,126,127]. Elevated AFP-L3 levels have been associated with a higher risk of HCC recurrence and early recurrences following tumor resection [128]. In other observational studies, the combination of AFP, AFP-L3, and DCP were associated with lower overall survival, larger tumors, and vascular invasion [97]. The prevalence of each tumor marker in that study showed that 23.2% of the cohort did not have any positive biomarker (cut-offs were for AFP >20 ng/mL, AFP-L3 >10% and DCP >40 mAU/mL), 32.1% had at least one positive biomarker, 22.3% 2 out of 3 and 22.3% 3 positive biomarkers [97]. In another cohort study evaluating the prognostic effect of combined biomarkers following liver resection, the recurrence-free survival (RFS) at 2-years of follow up was significantly lower with the presence of 1, 2 or 3 biomarkers including AFP, AFP-L3 and DCP (55% vs. 38% vs. 19%) [129]. There was an increasing prevalence of microvascular invasion or poorly differentiated tumors [129]. An elevated fraction of AFP-L3 has been linked with poor prognosis in surgically resected patients with HCC in Japan; associated with poorly differentiated tumors but showing less specificity for vascular invasion when compared to DCP [66]. The highly-sensitive AFP-L3 with a 5% cut-off value was associated with lower overall survival, even in patients with AFP values less than 20 ng/mL [54], and a higher risk of HCC recurrence after hepatectomy [130,131].

Finally, in a North American cohort study, the highly sensitive AFP-L3 in combination with AFP and DCP serum levels were associated with a higher risk of HCC recurrence after LT [132]. Indeed, the authors proposed a novel selection tool for patients beyond Milan criteria [133] using different cutoffs of AFP (>250 ng/mL), AFP-L3 (>35%) and DCP (>7.5 ng/mL) [132]. The inclusion of each biomarker with Milan criteria as a novel selection criterion showed that corresponding AUROCs were for Milan + AFP 0.68 (CI 0.60–0.76), AFP-L3 0.70 (CI 0.62–0.78) and DCP 0.70 (CI 0.62–0.78). However, a significant selection bias was observed due to the exclusive inclusion of HCC patients without HCC recurrence after LT; those without recurrence were discarded for blood samples storage and analysis. Thus, the association between these biomarkers and post LT HCC recurrence was biased in this study.

#### 3.1.3. DCP

DCP or PIVCA-II has also been explored as a prognostic marker in HCC. It has been widely associated with larger tumors, poor differentiation, and vascular invasion [127,134]. It has also been shown that its serum levels are higher after tumor hypoxia, and has been proposed as a predictive biomarker after anti-angiogenic therapies [135]. However, its prognostic value has been evaluated with different cut-off values [127]. The most commonly used was >40 mAU/mL either alone [127] or in combination with other biomarkers [129,136]. It was specifically associated with vascular invasion compared to AFP or AFP-L3 [68].

In regard with clinical outcomes, DCP has been associated with lower survival and a higher risk of HCC recurrence following liver resection, with higher specificity (92% vs. 87%; *P* < 0.001) and sensitivity (74% vs. 41%) compared to AFP values [137,138]. On the contrary, it has been associated with a lower discrimination power for HCC recurrence after living donor LT (LDLT) in Japan [139]. The AUROC for AFP, DCP and the neutrophil/lymphocyte ratio were 0.88, 0.76 and 0.62, respectively [139]. In this study, the authors proposed a new selection criteria for LDLT with a scoring model including the number of nodules (less or equal than 5), largest tumor diameter less or equal than 50 mm and the presence/absence of AFP >250 ng/mL, DCP >450 mAU/mL (LDLT Tokio University criteria) [139]. With the presence of at least 2 or 3 variables, the disease-free survival (DFS) and overall survival at 5-years following LT were 20% and 20%, respectively. Similarly, other authors proposed the Kyoto LDLT criteria incorporating DCP >400 mAU/mL, independently associated with HCC recurrence after LT [58]. The new Kyoto criteria included tumor number less or equal than 10, the largest nodule diameter less or equal than 50 mm, and DCP >450 mAU/mL. The AUROC was 0.84 compared with AFP [58]. Other authors have addressed lower cut-offs for HCC recurrence following LT (<300 mAU/mL) [140] or microvascular invasion in early HCC (>90 mAU/mL) [141] (Table 4).

#### 3.1.4. Osteopontin

This tumor biomarker has been associated with pathological features of aggressive HCC, including dedifferentiation and vascular invasion [72].

#### 3.1.5. Other Novel Biomarkers

Another promising biomarker associated with worse prognosis in early and advanced HCC stages was the HGF, known to promote tumor growth and metastasis. Encoded by the MET gene, HGF tyrosine kinase receptor inhibition was proposed as a potential therapeutic target [142]. The MET proto-oncogene encodes the tyrosine kinase receptor for the HGF, which activates downstream mechanisms upon tumor proliferation, invasion, and anti-apoptotic signals. MET overexpression in tumor tissue samples was associated with worse prognosis in patients receiving sorafenib. Indeed, expression of HGF/MET was observed to be increased after sorafenib exposure, suggesting an oncogenic escape mechanism [143]. However, a trial evaluating the effect of tivantinib, a MET pathway inhibitor, in patients highly expressing MET following sorafenib exposure, did not show any survival benefit compared to placebo [144]. Another MET inhibitor has recently shown efficacy for second-line treatment in advanced HCC [145]. Other angiogenic pathways have been associated with poor prognosis, including VEGF and ANG-2, among others, associated with vascular invasion, advanced tumor stages, lower DFS, and overall survival [146]. Micro-RNAs have also been linked to worst clinical outcomes in different observational studies [147] and are now being analyzed in large gene expression databases [148]. The neutrophil/leucocyte ratio (NLR) has been associated with worst overall survival in patients treated with sorafenib for advanced HCC [107] but it has not clearly shown to be associated with clinical outcomes in other clinical settings [139], although linked to HCC recurrence in other studies [149].

## 4. Serum and Tissue Biomarkers in HCC for Response Assessment Following Tumor Treatment

Clinical research has focused on the predictive capacity of each biomarker to evaluate tumor response before or after specific treatments. Apart from its prognostic association already described and their potential role in candidate selection, some biomarkers have been associated with a better response after treatment for early, intermediate, or advanced HCC. However, only AFP has been assessed as a selection tool for better candidates for liver transplantation [103,105,106,150] or a specific anti-angiogenic therapy [6].

One of the main proposed explanations is that HCC is a very heterogeneous tumor from a genetic standpoint with different oncogenic pathways [80]. The other hypothesis is that a higher expression of specific oncogenic pathways may not correlate with better clinical outcomes or treatment response. Moreover, although some biomarkers have been established as predictive factors for specific treatment in other cancers, in HCC this has not been the case. Immunotherapy based on checkpoint inhibitors in HCC has not been associated with increasing or decreasing efficacy based on PD-1 or PD-L1 assessment [3,5,151]. Previously, several directed-molecular therapies have failed to show any survival treatment benefit. One example was the high expression of c-MET on tumor tissue that showed promising results in a phase II trial [152], but did not show a survival benefit in second-line systemic treatment for advanced HCC [144].

### 4.1. Predictive Serum Biomarkers Following Locoregional Treatment

Decreasing levels of serum biomarkers might serve as a potential predictive marker of better outcomes. Indeed, AFP-L3 dynamic changes before or after locoregional HCC therapy or surgical resection has been associated with better overall survival in some observational studies [126,153]. PIVKA or DCP has also been associated with better outcomes as prognostic factor but it was not identified as a predictive factor of treatment response assessment. Indeed, DCP serum level at cut-off of less than 40 mAU/mL has been associated with better survival and lower recurrence rate in an HBV cohort after radiofrequency ablation [154]. VEGF serum level is a baseline prognostic factor following trans-arterial chemoembolization (TACE) [146]. This observation led to design a phase III trial evaluating the combination of TACE with sorafenib [155,156]. However, this biomarker did not correlate with better treatment response.

### 4.2. Predictive Serum Biomarkers Following Systemic Treatment

In the last years, systemic therapy for advanced HCC has improved the prognosis due to newly available drugs for first- and second-line systemic treatment options. During the last decades, further knowledge of HCC molecular mechanisms has led to the development of effective systemic treatment including tyrosine kinase inhibitors (TKIs) and immunotherapy. Sorafenib, the first antiangiogenic agent to be approved for first-line systemic treatment for advanced HCC, has shown to be effective irrespectively from disease etiology and baseline tumor biomarkers, including AFP, VEGF or ANG-2 [157,158]. However, no identified biomarker has been able to select appropriately the best candidates for any specific therapy; except from AFP serum levels and ramucirumab in second-line systemic therapy for advanced HCC [6]. Serum AFP, VEGF, and ANG-2 baseline levels were shown to be prognostic factors but were not predictive factors for sorafenib efficacy [107,159]. High baseline c-KIT and low baseline HGF were predictive factors of sorafenib treatment efficacy on univariate analysis and had a trend to be independently associated after adjustment for other prognostic factors [159].

A novel clinical term coined 10 years ago but yet not implemented in the daily practice, is the so-called “AFP response” [160]. This concept is the decreasing levels of AFP associated with better outcomes, but it was not identified as a predictive marker of better response [161]. A decreased in more than 50% from baseline levels of AFP following trans-arterial chemoembolization or trans-arterial radioembolization, defined as “AFP responders“ has been associated with better overall survival (HR 2.7 (CI 1.6–4.6) for mortality); however, the effect was not independent of that observed and evaluated through radiological imaging criteria [161,162]. The novel approach for “AFP response” might have a clinical role in patients under TKI, in which tumor shrinkage or a partial or complete radiological response is an infrequent event [117,118]. Another clinical definition of “AFP response” after systemic treatment might be a 20% decrease from baseline during the first 2–4 weeks of treatment initiation [160,163] or more than 50% at the end of the first month [164]. On the contrary, DCP has also been assessed following sorafenib, and a 2-fold increase in its serum levels might be a predictive factor of better response [165]. This preliminary observation was not further validated. However, there might have been several selection biases when assessing the “AFP response” as a prognostic variable. In some of these retrospective cohort studies, AFP baseline values in the group of “AFP responders” were significantly lower when compared to non-responders. The appropriate comparison should have been made in patients with baseline AFP levels >200 ng/mL [107,157].

Other predictive factors have been tested more recently in first and second-line systemic treatment for advanced HCC. Lenvatinib has been approved as a first-line systemic treatment option to sorafenib in an open-labeled phase 3 randomized clinical trial (REFLECT) [123]. In this trial, baseline VEGF, ANG-2, and fibroblast growth factor-2 (FGF-2) serum levels were associated with better overall survival with lenvatinib but not with sorafenib [166]. Besides, increasing VEGF levels following lenvatinib initiation was associated with better overall survival [166]. Tumor markers response and assessment through imaging data were evaluated in another retrospective cohort study including pre and post AFP or DCP levels at weeks 2 and 4 from lenvatinib initiation and correlated with the modified Response Evaluation Criteria for Solid Tumors (mRECIST) criteria [167,168]. In second-line systemic therapy, data coming from the RESORCE trial of regorafenib versus placebo including plasma and tissue samples showed that baseline serum levels of AFP and c-MET were associated with worst survival independently from regorafenib [125,169]. Nine micro-RNAs were associated with better overall survival in patients receiving regorafenib [169]. Thus, in this trial, baseline prognostic factors associated with survival were higher levels of AFP and c-MET, whereas predictive factors of better overall survival in patients receiving regorafenib were ANG-1, cystatin-B, the latency-associated peptide of transforming growth factor-beta 1 (LAPTGF ß1) and c-motif chemokine ligand 3 (MIP-1) [169]. Neither AFP nor c-MET were predictive factors of regorafenib benefit and its efficacy was independent of these biomarkers [125,169]. In another trial of sorafenib for prevention of HCC recurrence following liver resection or RFA, none of the tested biomarkers related to angiogenesis and proliferation, gene signatures, or mutations predicted sorafenib benefit over placebo [170].

### 4.3. Predictive Biomarkers Following Immunotherapy for HCC

Immunotherapy for cancer treatment has evolved into a complete novel paradigm. Cancer cells avoid lymphocyte T cell activation and proliferation, highly expressing “programmed cell death ligands” PD-1 ligand 1 and 2 (PD-L1, PD-L2) or “cytotoxic T lymphocyte protein 4” ligands (CTLA-4 and its ligands). The blockade of these unions through monoclonal antibodies (anti-PD-(L)1 or anti-CTLA-4) unblocks the anti-cancer immune host response (“immune checkpoint inhibitors”). The first anti-PD1 IgG4 monoclonal antibody available was nivolumab. Later, pembrolizumab (IgG4 anti-PD1), tremelimumab, and ipilimumab (IgG1 anti CTLA-4) and atezolizumab, durvalumab, and avelumab (IgG1 anti-PD-L1) appeared.

Recently, the CheckMate-459 phase III, an open-label randomized clinical trial (RCT) assessed nivolumab versus sorafenib in first-line therapy for patients with advanced HCC and did not show a survival benefit of nivolumab over sorafenib [HR 0.85 (CI 0.72-1.02); *P* = 0.07]. Additionally, there was not a significant difference in DFS, even showing a higher objective response rate (ORR) 15% vs. 7%. Nevertheless, nivolumab showed lower rate of grade 3/4 adverse events (22% vs. 49%) (NCT02576509).

More recently, another phase III, open-label RCT, IMbrave-150 (NCT03434379) [171] showed superiority of atezolizumab 1200 mg iv (anti-PD-L1) plus bevacizumab 15 mg/kg iv every 3 weeks versus sorafenib [HR 0.58 (CI 0.42–0.79); *P* = 0.0006]. Eligibility criteria included preserved liver function, systemic-naïve advanced HCC, ECOG 0-1 in the absence of main portal trunk invasion. Longer progression-free survival (PFS) with significantly higher ORR (27% vs. 12%) and DCR of 74% vs. 55% were observed. Similar incidence of all-grade adverse events and a lower incidence of grades 3/4 related adverse events were observed with the combination arm (36% vs. 46%). However, related severe adverse events and treatment discontinuation were higher in atezolizumab + bevacizumab arm (16% and 10%, respectively). Other combinations are being explored in phase III trials as first-line treatments: pembrolizumab + lenvatinib vs. lenvatinib (LEAP-002, NCT03713593), sorafenib vs. durvalumab (anti PDL-1) vs. durvalumab + tremelimumab (CTLA-4) (HIMALAYA, NCT03298451), cabozantinib with/without atezolizumab versus sorafenib (COSMIC-312; NCT03755791) and nivolumab + ipilimumab versus sorafenib or lenvatinib (CheckMate-9DW; NCT04039607).

In second-line systemic treatment, for either intolerant patients or under tumor progression after sorafenib, tremelimumab has been explored in an uncontrolled phase II trial in patients with HCV+ and was well tolerated [172]. Nivolumab was earlier explored in a phase I/II escalating and expansion cohorts, uncontrolled trial, the CheckMate-040, which included Child–Pugh A–B < 9 patients under progression with sorafenib (tolerant or intolerant) [3]. In this study, PD-L 1 expression on tumor biopsies (≥1% vs. <1%) was not associated with better survival but higher ORRs were observed (NCT02576509). Another phase III, double-blind RCT explored the treatment with pembrolizumab 200 mg iv every 3 weeks vs. placebo (KEYNOTE-240) in patients intolerant or under progression with sorafenib, without main portal trunk invasion and with AFP levels <400 ng/mL [173]. The trial was negative in terms of survival benefit according to its hypothesis test (13.9 vs. 10.6 months, HR of 0.78 (CI 0.71–0.99); *P* = 0.024). DFS did not reach its primary efficacy endpoint too. In a stratified analysis, patients with AFP < 200 ng/mL or with post-progression (vs. intolerant to sorafenib) or with HBV had the highest treatment benefit with pembrolizumab [173]. There is no robust data reporting the effect of PD-L1 tumor expression as a predictive marker of response to pembrolizumab. Data on PD-L1 tissue expression as a predictive tool with pembrolizumab was previously reported in a single center uncontrolled intervention study in 29 patients with advanced HCC who had developed disease progression or were intolerant to sorafenib [174]. In this study, serum pro and anti-inflammatory cytokines, as well as serum levels of PD-1 and PD-L1 and PD-L2, were measured at baseline. In only 10 patients there was tumor tissue available and only 4 were positive to PD-L1 staining, showing higher levels of plasma INF-y or IL-10 but PD-L1 staining was not correlated with plasma PD-L1 concentration. Plasma tumor-growth factor β (TGF-β) levels were associated with overall survival [174].

Unfortunately, overall radiological responses in patients receiving immunotherapy ranged from 15% to 20% in studies testing single-agent immunotherapy either in first and second-line settings [3,173]. Additionally, PD-L1 expression on tumor tissue, although it was associated with higher radiological responses, was not associated with a better survival benefit [3]. Thus, PD-L1 expression is not mandatory to select candidates for the treatment with these agents. On the contrary, patients with main portal trunk invasion or with high AFP values were excluded in these trials, as those with autoimmune hepatitis or other immunological disorders, thus making it impossible to assess their role in these scenarios.

Combining immunotherapy with anti-VEGF agents was explored to reduce VEGF-mediated immunosuppression in HCC [175,176]. In the IMbrave study, the ORR increased to 27% with a significant survival benefit over sorafenib [171]. However, until now, there are no predictive biomarkers to better select the appropriate candidates for immunotherapy. Some exploratory analyses were done in the phase 1b study of atezolizumab plus bevacizumab, showing that high expression of PD-L1 in tumor tissue, higher expression of VEGF receptor 2, and higher T-regulatory cells immune phenotype may be associated with better survival [177]. However, these biomarkers have not been yet analyzed in the IMbrave trial [171]. In this study, in patients with AFP values above 400 ng/mL, the combination arm did not show a significant benefit; still this should be cautiously analyzed [171]. A significant benefit was observed in patients with HBV related HCC [171].

## 5. Liquid Biopsy, Genomics and Other Biomarkers: The Future?

Finally, circulating free tumor deoxyribonucleic acid (ctDNA) in serum or plasma samples has been the focus of novel researches. However, this approach called “liquid biopsy” has several issues to be addressed [178,179,180]. Different forms of liquid biopsy have been described including circulating tumor cells, ctDNA, microRNA, and extracellular vesicles. First, circulating tumor cells have been assessed as predictors of treatment response in other tumors. However, the number of circulating tumor cells is challenging, specifically in early stages. Second, ctDNA is another type of liquid biopsy. However, it accounts for less than 1% of the total circulating cell-free DNA in normal conditions. Additionally, ctDNA may not reflect specific tumor DNA, on the contrary it may reflect necrotic or apoptotic tumor cells or cells that may not even originate from tumor tissue. Thus, the presence of cancer-specific genetic or epigenetic DNA changes is a key factor to identify in these circumstances. Moreover, these mutations maybe not specific for HCC in the context of cirrhosis, in which other DNA mutations or bacterial DNA may circulate in peripheral blood. Nevertheless, liquid biopsy is being researched not only for HCC early detection but also as a prognostic tool.

HCC is a heterogeneous tumor with substantial DNA mutations already reported. The most frequent are those in TERT (the gene that encodes the catalytic subunit of telomerase), TP53 (the p53 tumor suppressor gene), and CTNNB1 (the β-Catenin gene) [80,81,181]. These mutations depend on the underlying liver disease and are not completely present in most patients. TERT promoter mutation is most frequently reported in 60% of HCC cases; whereas TP53 and CTNNB1 in around 30%. Sampling error of tissue biopsy or tumor heterogeneity may explain discrepancies observed in circulating ctDNA mutations and those observed in tissue samples [180].

Genomic profiling of HCC has been developed during the last years. Two distinct classes have been proposed based on the genomic profile and its correlation with phenotypic profiles. The proliferation class has been associated with gene signatures of poor prognosis, TP53 mutations, and chromosomal instability. This proliferation class has been linked to HBV infection, poor cell differentiation, higher AFP values, and worse survival. On the contrary, the nonproliferation class has been associated with CTNNB1 mutations, immune exclusion, HCV and alcoholic liver disease, low tumor grade, lower frequency of vascular invasion, and better prognosis [80,81,182].

On the contrary, DNA methylation may be more specific. DNA methylation is an important mechanism of epigenetic regulation of gene expression and has been reported in HCC [183]. Whether these specific tissue-specific methylation patterns, either on liquid biopsy or tumor samples, are going to be useful biomarkers for early HCC diagnosis or prognosis is still uncertain and have not yet been embraced in clinical practice.

Advancement of medical technology may further identify novel gene expression signatures or new biomarkers such as those including ultimate sequencing identifying circulating or urine miRNAs, genomic diversity [80,81,182], and epigenetic factors [183]. However, until now, no specific genomic signature has been associated as a predictive marker of a better response to specific therapies for HCC.

Finally, radiomics in HCC is a novel but very preliminary approach based on artificial intelligence and imaging data [184,185,186]. Imaging features are registered on machine-learning algorithms, and these signs can assess and help HCC diagnosis and prognosis. However, radiomics is still a newborn technology-based approach that needs internal and external validation.

## 6. Conclusions

HCC surveillance and early diagnosis demand a continuous search for improvement in clinical tools for HCC screening due to its worldwide high associated mortality. The coexistence with chronic liver disease and inflammation has counterbalanced the accuracy of several tumor biomarkers, precluding them to be widely use in daily practice, except for AFP. Nevertheless, other tumor biomarkers have been developed and have shown to be associated with poor prognosis in different HCC stages and post-treatment assessment. Appropriate candidate selection for each therapeutic modality based solely on these biomarkers is still far away from its clinical applicability in the clinical decision-making processes. Ideal biomarkers for HCC are those that would enable clinicians to diagnose this cancer at asymptomatic stages and also, to help and identify better candidates in each tumor stage for appropriate therapeutic modalities [186,187]. So far, there is still a need for specific biomarkers to improve detection of HCC at early or very early stages, assess specific prognosis and prediction of treatment response.

## Figures and Tables

**Figure 1 cells-09-01370-f001:**
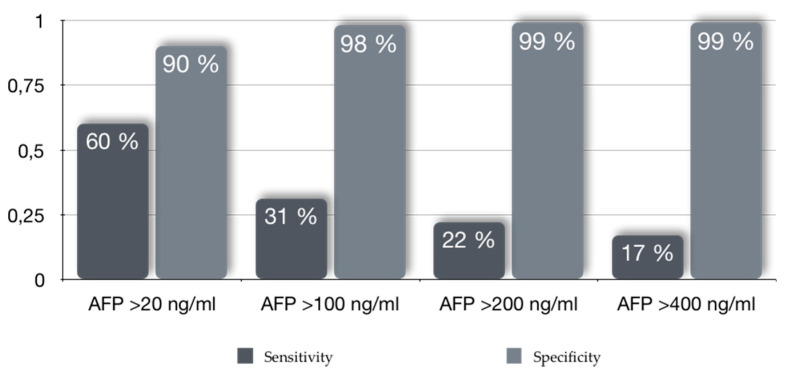
Sensitivities and specificities according to alpha-fetoprotein (AFP) cut-offs values for early hepatocellular carcinoma (HCC) diagnosis. ^1^ Adapted from [34].

**Table 1 cells-09-01370-t001:** Phases of biomarkers development and validation in cancer research.

Phase	Design of Study	Aims
I	Preclinical	Identify clinical biomarkers
II	Clinical—exploratory	Detection of disease ^1^
III	Observational—retrospective	Cancer detection at asymptomatic stages
IV	Observational—prospective	Extent and characteristics of the disease False referral rate
V	Trial—control	Impact on survival Tumor progression

^1^ Adapted from Early Detection Research Network (EDRN) of the National Cancer Institute from the United States of America [69,70].

**Table 2 cells-09-01370-t002:** Clinical staging algorithms for hepatocellular carcinoma (HCC) and inclusion of biomarkers.

Staging Algorithm	Clinical Variables Included	Tumor Variables	Hepatocellular Carcinoma (HCC) Biomarkers
BCLC [89,90]	ECOG Preserved liver function Portal hypertension	Number and diameters nodules Vascular invasion Extrahepatic spread	Not included
HKLC [96]	ECOG Child–Pugh	Number and diameters nodules Vascular invasion Extrahepatic spread	Not included
JIS [94,95]	Child–Pugh	TNM	Not included
CLIP [91]	Child–Pugh	Tumor extension >50% of liver volume	AFP > 400 ng/mL
GRETCH [92]	Karnofsky index Total bilirubin Alkaline phosphatase	Portal thrombosis	AFP > 35 ng/mL
CUPI [93]	Asymptomatic Ascites Total bilirubin Alkaline phosphatase	TNM staging	AFP > 500 ng/mL

**Table 3 cells-09-01370-t003:** Staging models for HCC prognosis including biomarkers.

Staging Algorithm	Clinical Variables	Tumor Imaging Features	Biochemical or HCC Biomarkers
BALAD score [99,100]	None	None	serum bilirubin, albumin, AFP > 400 ng/mL, AFP-L3 > 15% and DCP > 100 mAU/mL
ALBI grade [98]	None	None	serum bilirubin, albumin

**Table 4 cells-09-01370-t004:** Biomarkers used for liver transplantation selection criteria in patients with HCC.

LT Criteria	Tumor Imaging Features	Biomarkers	Expected Outcomes
AFP tumor volume [110]	Total tumor volume (TTV) >115 cm^3^	AFP >400 ng/mL	Overall survival <50% at 3 years
The AFP model [103]	Tumor number (1–3 vs. ≥4 nodules) Largest diameter (≤3 cm, 3–6 cm and >6 cm)	AFP ≤ 100 ng/mL, 101–1000 ng/mL and >1000 ng/mL	AFP score 2-point cut-off: 5-year survival and recurrence of 67.8% and 8.8%
Hanghzou criteria [111]	Sum of diameters (≤8 cm)	AFP > 400 ng/mL	Within Hanghzou: 5-year survival and recurrence: 70.8% and 35.7%.
Metroticket 2.0 [106]	Sum of nodules and largest diameter (Up-to-7)	Log10 AFP	Three different thresholds for HCC specific survival rate >50% at 5 years.
Tokio criteria [139]	Tumor number (≤5 nodules) Largest diameter (≤5 cm)	Beyond Tokio criteria AFP > 250 ng/mL DCP > 450 mAU/mL	2/3 criteria: 5-year survival 20%
Kyoto criteria [58]	Tumor number (≤10 nodules) Largest diameter (≤5 cm)	DCP > 400 mAU/mL	Beyond Milan & within Kyoto criteria: 5-year recurrence 4%
3-model biomarker approach [132]	Beyond Milan	AFP (>250 ng/mL) or AFP-L3 (>35%) or DCP (>7.5 ng/mL)	Higher recurrence with any of these criteria.

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
