# Peer review of "Biomarkers in Hepatocellular Carcinoma: Diagnosis, Prognosis and Treatment Response Assessment"

_cells, 2020, doi:10.3390/cells9061370_

Round 1
Reviewer 1 Report
Here is my review of the manuscript Title: Biomarkers in hepatocellular carcinoma: diagnosis, prognosis and treatment response assessment.
> 1. In general, it is a good review however major omission is present
> The latest treatment for HCC is immunotherapy. In fact, a PD-1 inhibitor combined with bevacizumab will soon be the first line standard of care replacing sorafenib.
> A recent randomized trial comparing the two treatments showed the PD-1 inhibitor/bevacizumab was superior to sorafenib. There is no mention of this clinical trial.
> In fact, a major omission is any mention or discussion of immunotherapy.
>
> 2. If immunotherapy is included in this paper, then the authors should discuss biomarkers for response to immunotherapy in HCC.
> For example, is PD-1 tumor expression important in predicting response or for prognosis in general?
> References that can be discussed include:
> El-Khoueiry in Lancet 2017,
> Feun in Cancer 2019
>
> This manuscript could be acceptable with major additions added above.
>
Author Response
Reviewer(s)' Comments to Author:
Reviewer 1:
Here is my review of the manuscript Title: Biomarkers in hepatocellular carcinoma: diagnosis, prognosis and treatment response assessment. In general, it is a good review however major omission is present
> The latest treatment for HCC is immunotherapy. In fact, a PD-1 inhibitor combined with bevacizumab will soon be the first line standard of care replacing sorafenib. A recent randomized trial comparing the two treatments showed the PD-1 inhibitor/bevacizumab was superior to sorafenib. There is no mention of this clinical trial.
Response: Yes, we agree. We now included the IMbrave trial as recently published (NEJM 2020). Please see page 13, last paragraph.
> In fact, a major omission is any mention or discussion of immunotherapy.
Response: As requested we specifically included this point in a new point including all trials evaluating immunotherapy in first and second-line (please see point 4.3. “Predictive biomarkers following immunotherapy for HCC”). Page 13, last paragraph.
> 2. If immunotherapy is included in this paper, then the authors should discuss biomarkers for response to immunotherapy in HCC. For example, is PD-1 tumor expression important in predicting response or for prognosis in general?
> References that can be discussed include:
> El-Khoueiry in Lancet 2017,
> Feun in Cancer 2019
Response: We thank for the reviewer´s suggestion. These references were previously cited (3,4,5,147). As suggested, we included another point (4.3) in which it is detailed the results of different trials with immunotherapy and its prognostic assessment with PD-L1 (please see pages 14 and 15).
> This manuscript could be acceptable with major additions added above.

Reviewer 2 Report
The authors wrote a comprehensive review on blood-based biomarkers in hepatocellular carcinoma for diagnosis, prognosis and treatment response assessment. The review is well structured and written overall. I have a few suggestions to improve the clarity of the review article. My comments are listed below.
Major comments
#1. There is various type of biomarkers. This review article specifically discussed blood-based biomarkers in HCC. This needed to be clarified in the title, abstract, and aim of the review article.
#2. The authors mainly focused on discussing traditional biomarker, namely AFP, DCP and AFP L3 while missing the section on novel liquid biopsy biomarker, particularly with regard to circulating tumor cells, ctDNA and EVs. There are a number of publications showing its utility in HCC surveillance, diagnosis, prognosis and treatment response assessment. Please refers to the following review articles that summarize literature. (PMID: 32017145; 31226727; 32289533)
#3. Please consider adding this citation under section 2.2.2.3. In this recently accepted paper (Clin Gastroenterol Hepatol. 2020 May 7. pii: S1542-3565), the authors developed and validated a model to identify patients who are at risk for HCC based on the change in serum level of AFP over time. The model could be used to assign patients to high-risk vs. low-risk groups, and might be used to select patients for surveillance should be further discussed.
#4. I would propose the author discuss GALAD score under the surveillance test section rather than tumor staging and prognosis. I also think the authors should discuss external validation of GALAD score as this study showed that GALAD is superior performance to US for early stage HCC detection, which may change the paradigm of HCC surveillance practice (PMID: 30464023).
#5. The reference list is not well organized. It needs additional work.
- Was refer 19 published in 2016? Please clarify what “Accepted Manuscript”means.
- Journal “Gastroenterology” was spelled differently in the reference list (e.g., Ref 42, 50). I noted similar issues with CGH and other journals.
- Name of authors appeared as initial (e,g, , reference 6). Please go over all reference use and appropriate format.
Minor comments
- Please rephrase “AFP differential diagnosis with benign conditions and other malignancies:” in line 93.
- Was there a particular reason why line 306-310 were italicized?
- line 510—needs a space between but and yet.
Author Response
Reviewer: 2
Comments to the Author
The authors wrote a comprehensive review on blood-based biomarkers in hepatocellular carcinoma for diagnosis, prognosis and treatment response assessment. The review is well structured and written overall. I have a few suggestions to improve the clarity of the review article. My comments are listed below.
Major comments
#1. There is various type of biomarkers. This review article specifically discussed blood-based biomarkers in HCC. This needed to be clarified in the title, abstract, and aim of the review article.
Response: We thank for the reviewer´s suggestion. However, not all these biomarkers are assessed in serum samples. In fact, MET-high vs low was done in tumor samples in the METIV-trial (already mentioned in page 11: “Other novel biomarkers”). As requested, we clarified this point in the abstract (lines 18, 20 and 22), introduction (lines 36 and 53), title of points 2.1, 2.2, 2.3, 2.4, 3, 4, 4.1, 4.2. Additionally, in point 4.3 immunotherapy was discussed and underlined that PD-L1 has been assessed in tissue samples (please see pages 14 and 15).
#2. The authors mainly focused on discussing traditional biomarker, namely AFP, DCP and AFP L3 while missing the section on novel liquid biopsy biomarker, particularly with regard to circulating tumor cells, ctDNA and EVs. There are a number of publications showing its utility in HCC surveillance, diagnosis, prognosis and treatment response assessment. Please refers to the following review articles that summarize literature. (PMID: 32017145; 31226727; 32289533)
Response: We thank for the reviewer´s suggestion. We included a specific section for “Liquid biopsy, genomics and other biomarkers: the future? (line 617). These three publications were included and reviewed, as recommended.
#3. Please consider adding this citation under section 2.2.2.3. In this recently accepted paper (Clin Gastroenterol Hepatol. 2020 May 7. pii: S1542-3565), the authors developed and validated a model to identify patients who are at risk for HCC based on the change in serum level of AFP over time. The model could be used to assign patients to high-risk vs. low-risk groups, and might be used to select patients for surveillance should be further discussed.
Response: We thank for the reviewer´s suggestion. We included a specific paragraph including the recommended article (line 160). However, this recently published paper, the predictive model included baseline variables associated with increased risk for disease progression (as a composite end-point). In this and other publications, presence of portal hypertension is an independent variable associated with HCC development. Thus, among patients with clinically significant portal hypertension, surveillance should be further stricter or underlined (line 160).
#4. I would propose the author discuss GALAD score under the surveillance test section rather than tumor staging and prognosis. I also think the authors should discuss external validation of GALAD score as this study showed that GALAD is superior performance to US for early stage HCC detection, which may change the paradigm of HCC surveillance practice (PMID: 30464023).
Response: We included as suggested the GALAD score in the surveillance section under a new point (2.5; line 260). Additionally, we included the suggested reference and discussed its results (line 165).
#5. The reference list is not well organized. It needs additional work.
Was refer 19 published in 2016? Please clarify what “Accepted Manuscript”means.
Response: This was an error using the automated citation program. It was corrected.
Journal “Gastroenterology” was spelled differently in the reference list (e.g., Ref 42, 50). I noted similar issues with CGH and other journals.
Response: This was an error using the automated citation program. All them now corrected.
Name of authors appeared as initial (e,g, , reference 6). Please go over all reference use and appropriate format.
Response: This was an error using the automated citation program. All of them were corrected.
Minor comments
- Please rephrase “AFP differential diagnosis with benign conditions and other malignancies:” in line 93.
Response: Agreed. We changed it (please see line 90).
- Was there a particular reason why line 306-310 were italicized?
Response: This was a typo error; we corrected it.
- line 510—needs a space between but and yet.
Response: This was a typo error; we corrected it.

Reviewer 3 Report
I reviewed with interest the manuscript cells-809719 by Pinero and colleagues who review the literature on hepatocellular carcinoma (HCC) biomarkers.
The review is divided into 5 sections: 1. General introduction; 2. Biomarkers for hepatocellular carcinoma surveillance and diagnosis; 3. Biomarkers in HCC for tumor staging and prognosis; 4. Biomarkers in HCC for response assessment following tumor treatment; 5. Conclusions.
Overall, the review is rather well written. The review mainly focuses on secreted circulating biomarkers, and mainly discusses on well-established biomarkers such as a-fetoprotein (AFP), AFP-L3 and DCP. The title should probably reflect this restrictive aspect.
Over the last decades, genomic analysis identified important biomarkers in HCC, either circulating (what about Glypican 3 (GPC3)?), or within the tissues.
The review should also describe the clinical relevance of gene expression signatures (some of them with documented prognostic values), as well as genomic alterations (e.g. mutations of TERT promoter as early alterations).
Imaging is also considered as relevant in terms of biomarker – what about radiomics as emerging tools?
What about extracellular vesicles? Circulating tumor cells (CTCs)? Circulating RNA, including microRNAs?
These emerging biomarkers are mentioned in the conclusive section. Given that experimental evidence exists, a specific section should be included in the review.
Minor points
A title should be included for some sub-sections (e.g. 2.2.1.1., 2.2.1.2.)
Page 3, line 122. “Adapted” versus “Adopted” (tithe of Figure 1)
Author Response
Reviewer: 3
I reviewed with interest the manuscript cells-809719 by Pinero and colleagues who review the literature on hepatocellular carcinoma (HCC) biomarkers.
The review is divided into 5 sections: 1. General introduction; 2. Biomarkers for hepatocellular carcinoma surveillance and diagnosis; 3. Biomarkers in HCC for tumor staging and prognosis; 4. Biomarkers in HCC for response assessment following tumor treatment; 5. Conclusions.
Overall, the review is rather well written. The review mainly focuses on secreted circulating biomarkers, and mainly discusses on well-established biomarkers such as a-fetoprotein (AFP), AFP-L3 and DCP. The title should probably reflect this restrictive aspect.
Over the last decades, genomic analysis identified important biomarkers in HCC, either circulating (what about Glypican 3 (GPC3)?), or within the tissues.
Response: We thank for the reviewer´s suggestion. As requested, we add a paragraph focusing on Glipican-3 and HCC early detection, diagnosis and prognosis (line 296).
The review should also describe the clinical relevance of gene expression signatures (some of them with documented prognostic values), as well as genomic alterations (e.g. mutations of TERT promoter as early alterations).
Response: We included another paragraph focusing on genomic alterations, as suggested. Please see lines 617 to 654.
Imaging is also considered as relevant in terms of biomarker – what about radiomics as emerging tools?
Response: We included another paragraph focusing on radiomics. Please see line 655.
What about extracellular vesicles? Circulating tumor cells (CTCs)? Circulating RNA, including microRNAs? These emerging biomarkers are mentioned in the conclusive section. Given that experimental evidence exists, a specific section should be included in the review.
Response: As suggested, another section describing genomic biomarkers was included. Line 638. MicroRNA was previously described on lines 313-318.
Minor points
A title should be included for some sub-sections (e.g. 2.2.1.1., 2.2.1.2.)
Response: Agreed. We included two different titles.
Page 3, line 122. “Adapted” versus “Adopted” (tithe of Figure 1)
Response: Agreed. We have changed as requested.

Round 2
Reviewer 1 Report
the authors left out a recent paper
"Phase 2 study of pembrolizumab and circulating biomarkers to predict anticancer response in advance, unresectable
hepatocellular carcinoma " Cancer Oct 15, 2019 3603-3614 in their discussion on which biomarkers can predict response to
PD1 inhibitors. They should discuss this paper and other similar biomarkers
Author Response
Reviewer(s)' Comments to Author:
Reviewer 1:
the authors left out a recent paper
"Phase 2 study of pembrolizumab and circulating biomarkers to predict anticancer response in advance, unresectable hepatocellular carcinoma " Cancer Oct 15, 2019 3603-3614 in their discussion on which biomarkers can predict response to PD1 inhibitors. They should discuss this paper and other similar biomarkers
Response: We now included this suggested article and some comments were written. Please see page 13, line 597. Reference number 174.

Reviewer 2 Report
Reference lists are overlapping (e.g., Ref 38,39 are the same). Please remove duplicate reference list in the reference manager and update them.
In table 1, phase 4 refers to "Observational – prospective". Please update.
Author Response
Reviewer 2:
Reference lists are overlapping (e.g., Ref 38,39 are the same). Please remove duplicate reference list in the reference manager and update them.
Response: We thank for the reviewer suggestion. This was amended as suggested. Line 815.
In table 1, phase 4 refers to "Observational – prospective". Please update.
Response: This was amended as suggested. Line 256.

Reviewer 3 Report
I reviewed with interest the revised version of manuscript cells-809719 by Pinero and colleagues entitled Biomarkers in hepatocellular carcinoma: diagnosis, prognosis and treatment response assessment. In this revised version, major and minor comments raised by the reviewers have been addressed.
The manuscript has been improved accordingly and now provides a great and updated overview of current biomarkers in hepatocellular carcinoma.
Author Response
Reviewer 3:
I reviewed with interest the revised version of manuscript cells-809719 by Pinero and colleagues entitled Biomarkers in hepatocellular carcinoma: diagnosis, prognosis and treatment response assessment. In this revised version, major and minor comments raised by the reviewers have been addressed.
The manuscript has been improved accordingly and now provides a great and updated overview of current biomarkers in hepatocellular carcinoma.
Response: We thank the reviewer’s comment.
